# Comparison between Go-GutDx, a novel diagnostic stool test kit with potential impact in low-income countries, and BioFire test

**Quarshie Glover**◉*, Xiao Jiang, Alexis Marie Onderak, Abigail Mapes◉,
Fauzia Hollnagel◉, Joseph Buckley, Chang Hee Kim, Dawd Siraj

Department of Medicine, University of Wisconsin Hospitals and Clinics, Madison, Wisconsin, United States of America

* gq.quarshie@gmail.com

## Abstract

### Introduction

Infectious diarrheal diseases are one of the leading causes of worldwide morbidity and mortality. The incidence of diarrhea is higher in Low-Middle-Income Countries (LMIC), where more than 90% of deaths from diarrheal diseases occur. Diagnostic tests for infectious diarrhea are not readily available in Low-Middle-Income Countries. Our study evaluates a novel, cost-effective, easy-to-use DNA stool testing kit for infectious diarrhea that can easily be rolled out in low-resource settings and has comparable performance to current testing modalities in the USA in terms of diagnostic utility.

### Methods

435 stool samples were tested using the novel stool testing Kit (Go-GutDx®). The stool samples were in groups of 8 and were tested using floating microspheres DNA extraction followed by Recombinase Polymerase Amplification (RPA) and lateral flow assay detection. Pathogens tested include *Clostridium difficile, Campylobacter jejuni, Salmonella enterica Typhimurium, Shigella spp., STEC (stx1, stx2), Vibrio spp.,* and *Yersinia enterocolitica*. The same samples were tested using BioFire GI Panel (gold standard), and the results were compared. Descriptive analysis was summarized as raw counts and frequencies and compared using Fisher's exact test. We conducted specificity and sensitivity analysis of the two diagnostic tests using the diagnostic testing package in STATA. Graphical illustrations were conducted using two-way line graphs with custom margins and axis. All analyses with p-values ≤ 0.05 were significant. All analyses were conducted using STATA version 17.

### Results

Results showed a sensitivity of 56.1% for *C. difficile*, and 58.6% for *Campylobacter jejuni*. 46.1% for *STEC (stx1, stx2)*, 83.3% for *Salmonella*, 0% for *Yersinia enterocolitica*, 66.7% for Vibrio spp and 65.0% for *Shigella spp*. Specificities were > 97% for all pathogens.

**Data availability statement:** All relevant data are within the manuscript and its Supporting Information files.

**Funding:** "Research reported in this publication was supported by the National Center For Advancing Translational Sciences of the National Institutes of Health under Award Number R44TR001912. The content is solely the responsibility of the authors and does not necessarily represent the official views of the National Institutes of Health." The funders had no role in study design, data collection and analysis, decision to publish, or preparation of the manuscript.

**Competing interests:** I have read the journal's policy and the authors of this manuscript have the following competing interests: [Chang Hee Kim is a stockholder and patent holder at GoDx and Xiao Jiang is a patent holder at GoDx disclose potential conflicts of interest]. This does not alter our adherence to PLOS ONE policies on sharing data and materials.

The positive predictive value was highest for *Campylobacter jejuni* 94.4%, followed by *C. difficile* 93.5%, *Salmonella* 86.2%, *Shigella* 76.5%, *STEC (stx1, stx2)* 50%, *Vibrio spp* 40%, and *Yersinia* 0%. The negative predictive value was 75.8% for *C. difficile*, with all other pathogens being above 98.3%. Table 3. The concordance between BioFire and Go-GutDx® for *C. difficile* testing was 80%. All other organisms showed a concordance of greater than 97% Table 2.

## Discussion

Our study confirms that Go-GutDx® is a novel diagnostic tool for diarrheal pathogens with statistically comparable sensitivity, specificity, positive and negative predictive values with the current gold standard testing kit, BioFire. In addition, the simplicity of the technology and lower price both to healthcare systems and to patients makes Go-GutDx® an ideal diagnostic test to be implemented in low- and middle-income countries.

## Introduction

Infectious diarrheal diseases are one of the leading causes of worldwide morbidity and mortality. [1] The incidence of diarrhea is higher in developing countries, where more than 90% of deaths from diarrheal diseases occur. [2] In Low or Middle-Income Countries (LMIC), diarrhea has a huge impact on long-term disability due to repeated early childhood infections. [1] Poor sanitation and infrastructure have been implicated in the increased number of cases. [3] Diagnostic testing of causative organisms for diarrhea plays an important role in decreasing disease burden and death associated with infectious diarrhea. Microbiologic diagnosis of diarrhea etiology is essential for providing targeted antibiotic therapy when indicated. [4] Diagnostic testing also helps in disease surveillance and identifying outbreaks, and this is a useful tool for implementing public health interventions. [1] In most resource-limited settings, diagnostic investigations to isolate organisms are limited. Thus, infectious diarrhea is treated empirically with antibiotics. [5] Antibiotic therapy is contraindicated in some cases of diarrhea caused by organisms like *Escherichia coli O157* and other strains producing Shiga-like toxins. [6] Empiric treatment can lead to unnecessary antibiotic treatment, exposing patients to adverse side effects of antibiotics and straining resources, as antibiotics are costly and access is limited in some developing countries. [1,7] Moreover, unnecessary treatment of diarrhea with antibiotics increases antibiotic resistance, which has health and economic implications.[8,9]

Due to the importance of diagnostic tests in evaluating and managing infectious diarrhea, it is imperative that testing modalities are readily available. However, this is not the case in low-income countries due to the cost, technical expertise, and time that is required to use current diagnostic modalities, which are easily accessible in developed countries. BioFire is an example of a diagnostic stool test that is widely used in the United States but not readily available in developing countries. Our study evaluates a novel, cost-effective, easy-to-use stool testing kit (Go-GutDx®) for infectious diarrhea that can easily be rolled out in low-resource settings and has comparable performance to BioFire in terms of diagnostic utility.

## Materials and Methods

Stool samples (n = 435) were collected from participants with diarrhea from two sites, the University of Texas Medical Branch (UTMB) and Hennepin Healthcare (HHR). The collected stool samples were either stored in Cary Blair media or raw and were tested using the NIH

Standard of Care, BioFire Film Array GI Pathogen Panel (FDA approved PCR based pathogen diagnostic test) by the Department of Laboratory Medicine (DLM). Samples were shipped to Go-GutDx®, Inc. on dry ice and stored at -80° C until testing. The stool samples were tested at Go-GutDx®, Inc. in groups of 8 using floating microspheres DNA extraction followed by RPA amplification and lateral flow assay detection. Go-GutDx® test has been designed for the detection of the most common diarrhea-causing pathogens. Of interest for this study are: *Clostridium difficile, Campylobacter jejuni, Salmonella enterica Typhimurium, Shigella spp., STEC (stx1, stx2), Vibrio spp.,* and *Yersinia enterocolitica.* After the detection was completed by Go-GutDx®, the results were compared with the BioFire results to calculate sensitivities specificities and positive and negative predictive values. Participants did not receive results from Go-GutDx®. The results obtained by BioFire were considered the "true results" of the clinical samples tested.

## Statistical Methodology

Descriptive analysis was conducted to summarize data as raw counts and frequencies and compared using a Fisher's exact test. We conducted specificity and sensitivity analysis of the two diagnostic tests using the diagnostic testing package in STATA. We also conducted percent agreement using an overall concordance between both tests. Graphical illustrations were conducted using two-way line graphs with custom margins and axis. All analyses with p-values ≤ 0.05 were considered to be significant. All analyses were conducted using STATA version 17.

## Results

Out of the 435 stool samples, 180 (41.4%) tested positive for *C. difficile*, 29 (6.7%) tested positive for *Campylobacter*, 13 (3.0%) for *STEC*, 30 (6.9%) for *Salmonella*, 5 (1.1%) for *Yersinia,* 3 (0.70%) for *Vibrio* and 20(4.6%) for *Shigella,* using BioFire. For the number of positive test results, using Go-GutDx® for each organism, results show that 108 (24.8%) tested positive for *C. difficile*, 18 (4.1%) for *Campylobacter,* 12 (2.8%) for STEC, 29 (6.7%) for *Salmonella,* 3 (0.70%) for *Yersinia,* 5 (1.1%) for *Vibrio,* and 17 (3.9%) for *Shigella*, Table 1. The concordance between BioFire and Go-GutDx® for *C. difficile* testing was 80%. All other organisms showed a concordance of greater than 97%, Table 2. The sensitivity, specificity, positive and negative predictive values for each detectable pathogen are shown in Table 3 **and** Fig 1. These indicate a sensitivity of 56.1% for *C. difficile*, 58.6% for *Campylobacter jejuni.* 46.1% for *STEC (stx1, stx2)*, 83.3% for *Salmonella,* 0% for *Yersinia enterocolitica*, 66.7% for *Vibrio spp* and 65.0% for *Shigella spp.* Specificities were > 97% for all pathogens. The positive predictive value was highest for *Campylobacter jejuni* 94.4%, followed by *C. difficile* 93.5%, *Salmonella* 86.2%, *Shigella* 76.5%, *STEC (stx1, stx2)* 50%, *Vibrio spp* 40%, and *Yersinia 0%.* The negative predictive value was 75.8% for *C. difficile*, with all other pathogens being above 98.3%.

## Discussion

Infectious diarrhea continues to be a major cause of morbidity and mortality, especially in developing countries where more diarrheal deaths occur. Diarrheal illnesses have major socio-economic implications and have led to long-term disability. Diagnostic investigations for managing diarrhea remain at the core of treatment and prevention. BioFire is the gold-standard diagnostic stool test for detecting microorganisms and is widely used in the United States. It has been shown to have great utility in the clinical setting. However, BioFire is expensive, requires a high level of technical expertise, and is more time-consuming and cumbersome to perform. Due to these limitations, it is not readily available in Low or

**Table 1. Descriptive summaries.**

|  | BioFire | Go-GutDx® | P |
|---|---|---|---|
| Clostridium difficile |  |  |  |
| Positive | 180 (41.4) | 108 (24.8) | <0.001[*] |
| Negative | 255 (58.6) | 327 (75.2) |  |
| Campylobacter |  |  |  |
| Positive | 29 (6.7) | 18 (4.1) | 0.13 |
| Negative | 406 (93.3) | 417 (95.9) |  |
| STEC |  |  |  |
| Positive | 13 (3.0) | 12 (2.8) | 1.0 |
| Negative | 422 (97.0) | 423 (97.2) |  |
| Salmonella |  |  |  |
| Positive | 30 (6.9) | 29 (6.7) | 1.0 |
| Negative | 405 (93.1) | 406 (93.3) |  |
| Yersinia |  |  |  |
| Positive | 5 (1.1) | 3 (0.70) | 0.73 |
| Negative | 430 (98.9) | 432 (99.3) |  |
| Vibrio |  |  |  |
| Positive | 3 (0.70) | 5 (1.1) | 0.73 |
| Negative | 432 (99.3) | 430 (98.9) |  |
| Shigella |  |  |  |
| Positive | 20 (4.6) | 17 (3.9) | 0.74 |
| Negative | 415 (95.4) | 418 (96.1) |  |

[*]Statistically significant.

**Table 2. Concordance between the two tests.**

|  | Agreement (%) | 95% CI |
|---|---|---|
| Clostridium difficile | 80 | 76.5 – 83.9 |
| Campylobacter | 97.0 | 95.4 – 98.6 |
| STEC | 97.4 | 95.4 – 98.6 |
| Salmonella | 98.0 | 96.5 – 99.2 |
| Yersinia | 98.2 | 96.9 – 99.4 |
| Vibrio | 99.1 | 98.2 – 99.9 |
| Shigella | 97.5 | 95.9 – 98.9 |

Middle-Income Countries. One sample of BioFire costs the lab approximately $160 to buy, and in the US, labs charge around $1600 for one sample to the patient. BioFire instrument costs $40,000. BioFire test takes 1.2 hours to run, and performing the test requires training on how to operate a complex instrument. Our study introduces a novel point-of-care DNA stool testing kit (Go-GutDx®) for infectious diarrhea that has promising results, performs comparatively to BioFire, and has great potential to be used in low-resource settings due to low cost and ease of use. The reagent cost per sample of Go-GutDx® is roughly $11.56. With large-scale manufacturing, we estimate the cost should be under $5. It takes 30-40 minutes per sample to perform the test. It is easy to perform because it only requires pipetting skills and there is no instrument to learn to operate and maintain.

**Table 3. Sensitivity, specificity, and predictive values.**

| | Go-GutDx® | | | | | | |
|---|---|---|---|---|---|---|---|
| | Negative | Positive | Pr | Se (95% CI) | Sp (95% CI) | PPV (95% CI) | NPV (95% CI) |
| BioFire | | | | | | | |
| A: C. difficile | | | | | | | |
| Negative | 248 | 7 | 24.8% | 56.1 (51.4 – 60.8) | 97.2 (95.7 – 98.8) | 93.5 (91.2 – 95.8) | 75.8 (71.8 – 79.9) |
| Positive | 79 | 101 | | | | | |
| B: Campylobacter | | | | | | | |
| Negative | 405 | 1 | 4.1% | 58.6 (54.0 – 63.2) | 99.8 (99.3 – 100) | 94.4 (92.3 – 96.6) | 97.1 (95.6 – 98.7) |
| Positive | 12 | 17 | | | | | |
| C: STEC | | | | | | | |
| Negative | 416 | 6 | 2.8% | 46.1 (41.5 – 50.8) | 98.6 (97.5 – 99.7) | 50.0 (45.3 – 54.7) | 98.3 (97.1 – 99.5) |
| Positive | 7 | 6 | | | | | |
| D: Salmonella | | | | | | | |
| Negative | 401 | 4 | 6.7% | 83.3 (79.8 – 86.8) | 99.0 (98.1 – 99.9) | 86.2 (83.0 – 89.4) | 98.8 (97.7 – 99.8) |
| Positive | 5 | 25 | | | | | |
| E: Yersinia | | | | | | | |
| Negative | 427 | 3 | 0.7% | 0.0 (0.0 – 0.0) | 99.3 (98.5 – 100) | 0.0 (0.0 – 0.0) | 98.9 (97.8 – 99.9) |
| Positive | 5 | 0 | | | | | |
| F: Vibrio | | | | | | | |
| Negative | 429 | 3 | 1.1% | 66.7 (62.2- 71.1) | 99.3 (98.5 – 100) | 40 (35.4 – 44.6) | 99.8 (99.3 – 100) |
| Positive | 1 | 2 | | | | | |
| G: Shigella | | | | | | | |
| Negative | 411 | 4 | 3.9% | 65.0 (60.5 – 69.5) | 99.0 (98.1 – 99.9)) | 76.5 (72.5 – 80.5) | 98.3 (97.1 – 99.5) |
| Positive | 7 | 13 | | | | | |

Se = Sensitivity, Sp = Specificity, PPV = Positive Predictive Value, NPV = Negative predictive value, pr = Prevalence

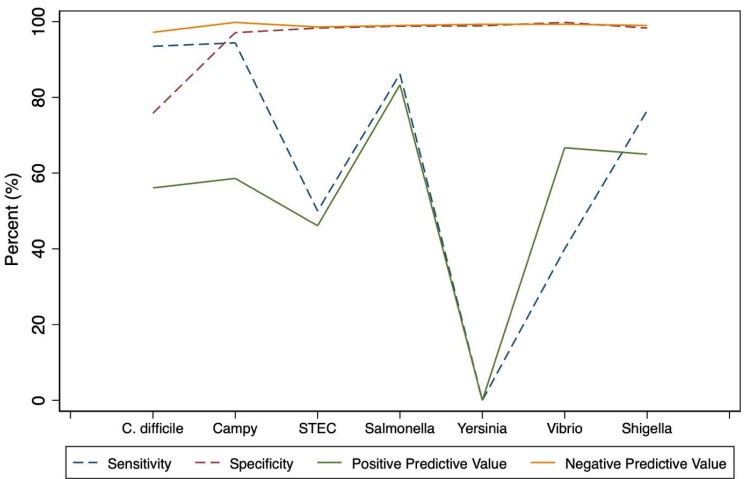

**Fig 1. Sensitivity, specificity, and predictive values.**

In our study, 435 stool samples were collected and tested with BioFire. The same stool samples were tested with the novel Go-GutDx® test. After the detection was completed by Go-GutDx®, the results were compared with the BioFire results to calculate sensitivities,

specificities, positive and negative predictive values, and concordance between the two testing modalities for 7 different pathogens. The results obtained by BioFire were considered the "true results" of the clinical samples tested. In analyzing the statistical outcomes of each organism, Go-GutDx® testing has performed reasonably well in comparison to the gold standard (BioFire), and balancing its cost-effectiveness and utility in low-resource settings, it can serve as a starting point for diagnosis. Several studies in LMIC have shown that the most common pathogens isolated from infective diarrhea include *E. coli, Salmonella, Shigella*, and *Vibrio*, with *E. coli* being the commonest bacteria pathogen[10–12], thus Go-GutDx® would be an effective diagnostic test for most cases of diarrhea. Go-GutDx® did not perform as well as BioFire at diagnosing *C. difficile*. Despite that, the concordance of Go-GutDx® with BioFire at detecting *C. difficile* was 80%. In many LMICs, *C. difficile* prevalence is unknown as testing is expensive and technologically demanding. The introduction of Go-GutDx® would be a promising step in studying the prevalence of the disease.

Of note, the results for *Vibrio* testing showed that Go-GutDx® performed better than BioFire in detecting *Vibrio* with 5 positive test results (1.1%) as compared to BioFire with 3 positive test results (0.70%). Even though this shows that Go-GutDx® may be as good as BioFire in detecting Vibrio, these positive results may be false positives and may be due to the quality of the test, which may clear out in a bigger study that is done in a population where the disease is much more prevalent. In general, the concordance between Go-GutDx® and BioFire in detecting various organisms is high, ranging from 80% to 99.1%, with more than 90% for six microorganisms. These results are encouraging and show that Go-GutDx® can perform comparably to BioFire as a stool testing kit.

One limitation of our study has been the overall low prevalence of GI pathogens in the samples tested. Considering this, a larger study in an endemic setting will be critical to validate the test sensitivity, specificity, and concordance of Go-GutDx® with BioFire. A larger study in an endemic area has the additional advantage of enhancing the performance of Go-GutDx®, as the higher prevalence of infection will influence statistical outcomes.

The negative predictive value for Go-GutDx® in detecting various microorganisms is impressive, ranging from 97.1% to 99.8% for all pathogens except C. difficile, which has a negative predictive value of 75.8%. This gives Go-GutDx® additional advantage in high-prevalence, low-income countries as a tool to rule out gut bacterial infections, thereby decreasing the empiric use of antibiotics. This, in turn, contributes to reducing antibiotic resistance and healthcare costs in LMICs.

## Conclusion

In conclusion, our study confirms that Go-GutDx® is a novel diagnostic tool for identification of diarrheal pathogens and it has statistically comparable sensitivity, specificity, positive and negative predictive values especially for *C.diff, Salmonella, Shigella,* and *Campylobacter,* with the current gold standard testing kit, BioFire. In addition, the simplicity of the technology and lower price makes Go-GutDx® an ideal diagnostic test in low- and middle-income countries.

AcknowledgmentWe are grateful for the entire team's contribution to this project.

## Author contributions

**Conceptualization:** Alexis Marie Onderak.

**Data curation:** Xiao Jiang.

**Formal analysis:** Xiao Jiang, Fauzia Hollnagel, Dawd Siraj.

**Investigation:** Xiao Jiang.

**Methodology:** Xiao Jiang, Alexis Marie Onderak, Fauzia Hollnagel.

**Project administration:** Abigail Mapes, Chang Hee Kim, Dawd Siraj.

**Supervision:** Quarshie Glover, Chang Hee Kim, Dawd Siraj.

**Writing – original draft:** Quarshie Glover, Fauzia Hollnagel, Chang Hee Kim, Dawd Siraj.

**Writing – review & editing:** Quarshie Glover, Xiao Jiang, Alexis Marie Onderak, Abigail Mapes, Fauzia Hollnagel, Joseph Buckley, Chang Hee Kim, Dawd Siraj.

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
