## [Decision Letter · Decision Letter 0]

9 Dec 2024

PONE-D-24-09347Comparison between Go-GutDx®, a novel diagnostic stool test kit with potential impact in low-income countries, and BioFire testPLOS ONE

Dear Dr. Glover,

Thank you for submitting your manuscript to PLOS ONE. After careful consideration, we feel that it has merit but does not fully meet PLOS ONE’s publication criteria as it currently stands. Therefore, we invite you to submit a revised version of the manuscript that addresses the points raised during the review process. Please go through the Reviewers comments carefully and draft your responses and suggested changes accordingly.

We look forward to receiving your revised manuscript.

Kind regards,

Furqan Kabir

Academic Editor

PLOS ONE

Journal Requirements:

3. Thank you for stating the following financial disclosure: [Research reported in this publication was supported by the National Center For Advancing Translational

Sciences of the National Institutes of Health under Award Number R44TR001912. The content is

solely the responsibility of the authors and does not necessarily represent the official views of

the National Institutes of Health.]. Please state what role the funders took in the study. If the funders had no role, please state: "The funders had no role in study design, data collection and analysis, decision to publish, or preparation of the manuscript." If this statement is not correct you must amend it as needed. Please include this amended Role of Funder statement in your cover letter; we will change the online submission form on your behalf.

4. Thank you for stating the following in the Competing Interests section: [I have read the journal's policy and the authors of this manuscript have the following competing interests: [Chang Hee Kim is a stockholder and patent holder at GoDx and Xiao Jiang is a patent holder at GoDx disclose potential conflicts of interest]]. Please confirm that this does not alter your adherence to all PLOS ONE policies on sharing data and materials, by including the following statement: "This does not alter our adherence to PLOS ONE policies on sharing data and materials.” (as detailed online in our guide for authors http://journals.plos.org/plosone/s/competing-interests). If there are restrictions on sharing of data and/or materials, please state these. Please note that we cannot proceed with consideration of your article until this information has been declared. Please include your updated Competing Interests statement in your cover letter; we will change the online submission form on your behalf.

5. We note that your Data Availability Statement is currently as follows: [All relevant data are within the manuscript and its Supporting Information files.] Please confirm at this time whether or not your submission contains all raw data required to replicate the results of your study. Authors must share the “minimal data set” for their submission. PLOS defines the minimal data set to consist of the data required to replicate all study findings reported in the article, as well as related metadata and methods (https://journals.plos.org/plosone/s/data-availability#loc-minimal-data-set-definition). For example, authors should submit the following data: - The values behind the means, standard deviations and other measures reported; - The values used to build graphs; - The points extracted from images for analysis. Authors do not need to submit their entire data set if only a portion of the data was used in the reported study. If your submission does not contain these data, please either upload them as Supporting Information files or deposit them to a stable, public repository and provide us with the relevant URLs, DOIs, or accession numbers. For a list of recommended repositories, please see https://journals.plos.org/plosone/s/recommended-repositories. If there are ethical or legal restrictions on sharing a de-identified data set, please explain them in detail (e.g., data contain potentially sensitive information, data are owned by a third-party organization, etc.) and who has imposed them (e.g., an ethics committee). Please also provide contact information for a data access committee, ethics committee, or other institutional body to which data requests may be sent. If data are owned by a third party, please indicate how others may request data access.

Reviewers' comments:

Reviewer's Responses to Questions

**Comments to the Author**

1. Is the manuscript technically sound, and do the data support the conclusions?

Reviewer #1: Yes

Reviewer #2: Yes

2. Has the statistical analysis been performed appropriately and rigorously? 

Reviewer #1: Yes

Reviewer #2: Yes

3. Have the authors made all data underlying the findings in their manuscript fully available?

Reviewer #1: Yes

Reviewer #2: No

4. Is the manuscript presented in an intelligible fashion and written in standard English?

Reviewer #1: Yes

Reviewer #2: No

5. Review Comments to the Author

Reviewer #1: It is a well-written paper about the comparison between a new diagnostic tool Go-GutDx with results available in 45 minutes and BioFire Test. I have no comment except the lack of interest with the figure which is not numbered.

Reviewer #2: Thank you for the opportunity to review this study. The research is interesting and addresses an important need in low-resource settings for a cost-effective and user-friendly diagnostic tool for diarrheal pathogens.

The comparison of the novel Go-GutDx® test with the current gold standard BioFire Film Array GI Pathogen Panel is valuable in understanding the performance of this new diagnostic method.

Certainly, here are more detailed suggestions to improve the study and manuscript:

Major comments:

1. Sample collection and participant demographics: Include a more detailed description of the study population, such as age groups, gender distribution, and any relevant clinical symptoms or underlying conditions. This information will help readers understand the context and generalizability of the results.

2. Sample storage and transportation: Clarify whether Cary Blair media was used for all samples and, if not, how the samples were otherwise stored and transported. It is crucial to ensure consistency in sample handling to minimize potential effects on test performance.

3. Go-GutDx® test details: Provide more information about the test's design and targets for each pathogen, including any potential cross-reactivity with other microorganisms. This information will help readers understand the test's strengths, limitations, and potential applicability.

4. Discussion of clinical implications: Expand on the practical implications of the results, particularly the lower sensitivity of Go-GutDx compared to BioFire. Discuss the potential impact on patient care and public health in the context of resource-limited settings.

Minor comments:

1. Non-standard abbreviations: Ensure that abbreviations are standardized throughout the manuscript. For example, "Go-GutDx" and "BioFire" should be written consistently with consistent capitalization.

2. Grammar and clarity: Review the manuscript for grammar, punctuation, and clarity. Some sentences may require revisions to improve readability.

3. Expanded comparison with BioFire: Consider discussing the cost, technical expertise, and time differences between Go-GutDx and BioFire in more detail. This information is important for understanding the potential benefits and limitations of Go-GutDx in low-resource settings.

4. Further validation: Discuss plans for further validation of Go-GutDx in larger studies and in low-resource settings, as this will be essential for establishing its clinical utility and cost-effectiveness in such contexts.

6. PLOS authors have the option to publish the peer review history of their article (what does this mean? ). If published, this will include your full peer review and any attached files.

**Do you want your identity to be public for this peer review?** For information about this choice, including consent withdrawal, please see our Privacy Policy .

Reviewer #1: No

Reviewer #2: **Yes: ** Dr. Ebrahim Kousari

---

## [Author Response · Author response to Decision Letter 1]

22 Jan 2025

Response To Reviewers

Thank you for the comments. Our protocol has also been submitted to this website https://www.protocols.io/file/smqpcki7f.pdf Figures have been uploaded to PACE as instructed https://pacev2.apexcovantage.com/ Each question is followed by an answer as seen below

Consent

Please provide additional details regarding participant consent. In the ethics statement in the Methods and online submission information, please ensure that you have specified (1) whether consent was informed and (2) what type you obtained (for instance, written or verbal, and if verbal, how it was documented and witnessed). If your study included minors, state whether you obtained consent from parents or guardians. (3) If the need for consent was waived by the ethics committee, please include this information.

The samples were de-identified, discarded samples collected for another purpose. Thus, no informed consent was obtained. Please refer to the attached study protocol.

The need for consent was waived by the IRB. Please see attached for the IRB exemptions.

Funding Statement

Thank you for stating the following financial disclosure: [Research reported in this publication was supported by the National Center For Advancing Translational Sciences of the National Institutes of Health under Award Number R44TR001912. The content is solely the responsibility of the authors and does not necessarily represent the official views of the National Institutes of Health.]. Please state what role the funders took in the study. If the funders had no role, please state: "The funders had no role in study design, data collection and analysis, decision to publish, or preparation of the manuscript." If this statement is not correct you must amend it as needed. Please include this amended Role of Funder statement in your cover letter; we will change the online submission form on your behalf.

Competing Interest

Thank you for stating the following in the Competing Interests section: [I have read the journal's policy and the authors of this manuscript have the following competing interests: [Chang Hee Kim is a stockholder and patent holder at GoDx and Xiao Jiang is a patent holder at GoDx disclose potential conflicts of interest]]. Please confirm that this does not alter your adherence to all PLOS ONE policies on sharing data and materials, by including the following statement: "This does not alter our adherence to PLOS ONE policies on sharing data and materials.” (as detailed online in our guide for authors http://journals.plos.org/plosone/s/competing-interests). If there are restrictions on sharing of data and/or materials, please state these. Please note that we cannot proceed with consideration of your article until this information has been declared. Please include your updated Competing Interests statement in your cover letter; we will change the online submission form on your behalf.

Chang Hee Kim is a stockholder and patent holder at GoDx and Xiao Jiang is a patent holder at GoDx to disclose potential conflicts of interest. This does not alter our adherence to PLOS ONE policies on sharing data and materials. There are no restrictions on sharing of data and/or materials.

Data Availability

We note that your Data Availability Statement is currently as follows: [All relevant data are within the manuscript and its Supporting Information files.] Please confirm at this time whether or not your submission contains all raw data required to replicate the results of your study. Authors must share the “minimal data set” for their submission.

PLOS defines the minimal data set to consist of the data required to replicate all study findings reported in the article, as well as related metadata and methods (https://journals.plos.org/plosone/s/data-availability#loc-minimal-data-set-definition). For example, authors should submit the following data: - The values behind the means, standard deviations and other measures reported; - The values used to build graphs; - The points extracted from images for analysis. Authors do not need to submit their entire data set if only a portion of the data was used in the reported study. If your submission does not contain these data, please either upload them as Supporting Information files or deposit them to a stable, public repository and provide us with the relevant URLs, DOIs, or accession numbers. For a list of recommended repositories, please see https://journals.plos.org/plosone/s/recommended-repositories. If there are ethical or legal restrictions on sharing a de-identified data set, please explain them in detail (e.g., data contain potentially sensitive information, data are owned by a third-party organization, etc.) and who has imposed them (e.g., an ethics committee). Please also provide contact information for a data access committee, ethics committee, or other institutional body to which data requests may be sent. If data are owned by a third party, please indicate how others may request data access.

We confirm that our submission contains all raw data required to replicate the results of our study.

Sample collection and participant demographics: Include a more detailed description of the study population, such as age groups, gender distribution, and any relevant clinical symptoms or underlying conditions. This information will help readers understand the context and generalizability of the results.

Thank you for this question. Per our study design and protocol, we do not need the demographic data, and we did not collect it. Please find attached study protocol. Whiles demographics may be helpful, our study focused more on answering the questions about the performance of a novel test kit against a standard test Kit in the same population of people. Stool samples from the same person were tested with both test Kits. So, this makes the demographic information irrelevant in answering our clinical question.

Sample storage and transportation: Clarify whether Cary Blair media was used for all samples and, if not, how the samples were otherwise stored and transported. It is crucial to ensure consistency in sample handling to minimize potential effects on test performance.

Initially, raw stools were saved and frozen at -80 deg C. Then, Cary-Blair was used for all samples to increase enrollment. The product will be made to use stool samples in Cary-Blair.

Go-GutDx® test details: Provide more information about the test's design and targets for each pathogen, including any potential cross-reactivity with other microorganisms. This information will help readers understand the test's strengths, limitations, and potential applicability.

Primers and targets are attached. Bioinformatic and analytical cross reactivity experiments were performed to limit cross-reactivity.

Discussion of clinical implications: Expand on the practical implications of the results, particularly the lower sensitivity of Go-GutDx compared to BioFire. Discuss the potential impact on patient care and public health in the context of resource-limited settings.

The sensitivity of Go-GutDx® was not lower than BioFire. In fact, the concordance for most GI pathogens was over 95%. The overall diagnosis of a pathogen was low in our sample specimens, considering this study was performed in a high-income, low-GI bacterial pathogen prevalence area. Despite that, our test performed comparably to BioFire. To validate the findings from this study, one of our recommendations is to repeat the study in a high-GI pathogen country with a larger sample size. This study in low middle income countries will be considered a validation of the new diagnostic kit.

Expanded comparison with BioFire: Consider discussing the cost, technical expertise, and time differences between Go-GutDx and BioFire in more detail. This information is important for understanding the potential benefits and limitations of Go-GutDx in low-resource settings.

We appreciate this comment and we have already added it. “BioFire is expensive, requires a high level of technical expertise, and is more time-consuming and cumbersome to perform. Due to these limitations, it is not readily available in Low or Middle-Income Countries. One sample of BioFire costs the lab approximately $160 to buy, and in the US, labs charge around $1600 for one sample to the patient. BioFire instrument costs $40,000. BioFire test takes 1.2 hours to run, and performing the test requires training on how to operate a complex instrument. Our study introduces a novel point-of-care DNA stool testing kit (Go-GutDx®) for infectious diarrhea that has promising results, performs comparatively to BioFire, and has great potential to be used in low-resource settings due to low cost and ease of use. The reagent costs per sample of Go-GutDx® is roughly $11.56. With large-scale manufacturing, we estimate the cost should be under $5. It takes 30-40 minutes per sample to perform the test. It is easy to perform because it only requires pipetting skills and there is no instrument to learn to operate and maintain”

Further validation: Discuss plans for further validation of Go-GutDx in larger studies and in low-resource settings, as this will be essential for establishing its clinical utility and cost-effectiveness in such contexts.

Thank you for your comments. We have re-written the discussion part of our manuscript emphasizing the importance of a validation study as the next step to introduce this new diagnostic kit to low middle income countries.

---

## [Editor Report · Decision Letter 1]

29 Jan 2025

Comparison between Go-GutDx®, a novel diagnostic stool test kit with potential impact in low-income countries, and BioFire test

PONE-D-24-09347R1

Dear Dr. Glover,

We’re pleased to inform you that your manuscript has been judged scientifically suitable for publication and will be formally accepted for publication once it meets all outstanding technical requirements.

Kind regards,

Furqan Kabir

Academic Editor

PLOS ONE
---

## [Editor Report · Acceptance letter]

PONE-D-24-09347R1

PLOS ONE

Dear Dr. Glover,

I'm pleased to inform you that your manuscript has been deemed suitable for publication in PLOS ONE. Congratulations! Your manuscript is now being handed over to our production team.

Kind regards,

on behalf of

Dr. Furqan Kabir

Academic Editor

PLOS ONE